# Expression levels of MHC class I molecules are inversely correlated with promiscuity of peptide binding

**Paul E Chappell[1][†], El Kahina Meziane[2][†], Michael Harrison[2], Łukasz Magiera[2], Clemens Hermann[2], Laura Mears[2], Antoni G Wrobel[2], Charlotte Durant[2], Lise Lotte Nielsen[3], Søren Buus[3], Nicola Ternette[4], William Mwangi[5], Colin Butter[5], Venugopal Nair[5], Trudy Ahyee[6], Richard Duggleby[6], Alejandro Madrigal[6,7], Pietro Roversi[1][‡], Susan M Lea[1]\*, Jim Kaufman[2,5,8,9]\***

[1]Sir William Dunn School of Pathology, University of Oxford, Oxford, United Kingdom; [2]Department of Pathology, University of Cambridge, Cambridge, United Kingdom; [3]Faculty of Health Sciences, University of Copenhagen, Copenhagen, Denmark; [4]Target Discovery Institute, University of Oxford, Oxford, United Kingdom; [5]Pirbright Institute, Compton, United Kingdom; [6]Anthony Nolan Research Institute, The Royal Free Hospital, London, United Kingdom; [7]University College London, London, United Kingdom; [8]Basel Institute for Immunology, Basel, Switzerland; [9]Department of Veterinary Medicine, University of Cambridge, Cambridge, United Kingdom

**\*For correspondence:** susan.lea@path.ox.ac.uk (SML); jfk31@cam.ac.uk (JK)

[†]These authors contributed equally to this work

**Present address:** [‡]Department of Biochemistry, University of Oxford, Oxford, United Kingdom

**Competing interests:** The authors declare that no competing interests exist.

**Abstract** Highly polymorphic major histocompatibility complex (MHC) molecules are at the heart of adaptive immune responses, playing crucial roles in many kinds of disease and in vaccination. We report that breadth of peptide presentation and level of cell surface expression of class I molecules are inversely correlated in both chickens and humans. This relationship correlates with protective responses against infectious pathogens including Marek's disease virus leading to lethal tumours in chickens and human immunodeficiency virus infection progressing to AIDS in humans. We propose that differences in peptide binding repertoire define two groups of MHC class I molecules strategically evolved as generalists and specialists for different modes of pathogen resistance. We suggest that differences in cell surface expression level ensure the development of optimal peripheral T cell responses. The inverse relationship of peptide repertoire and expression is evidently a fundamental property of MHC molecules, with ramifications extending beyond immunology and medicine to evolutionary biology and conservation.

## Introduction

Highly polymorphic class I molecules encoded by the major histocompatibility complex (MHC) are crucial in the adaptive immune response to viruses and some intracellular bacteria, binding peptides inside the cell and presenting them on the cell surface to CD8 T lymphocytes (*Blum et al., 2013*; *International HIV Controllers Study et al., 2013*; *Trowsdale and Knight, 2013*). The impact of the MHC in response to human immunodeficiency virus (HIV) is well recognized, with class I alleles like HLA-B\*35:01 leading to rapid onset of AIDS while HLA-B\*57:01 and HLA-B\*27:05 confer long-term non-progression (*Carrington et al., 1999*; *Goulder and Walker, 2012*; *International HIV Controllers Study et al., 2013*). Various explanations for these associations have been suggested, including antigen presentation of a particularly effective peptide or of a number of peptides to cytotoxic CD8 T cells or recognition (primarily) independent of peptide by natural killer (NK) cells (*Carrington et al., 1999*; *Martin et al., 2002*; *Kosmrlj et al., 2010*; *International HIV Controllers Study et al., 2013*).

**eLife digest** Our immune system has the remarkable ability to recognize and destroy damaged cells or invading microbes while leaving the healthy cells in the body alone. Groups of proteins called 'MHC class I molecules' play important roles in defence against invading microbes. If a cell becomes infected with a virus or a bacterium, these molecules can recognize and bind to fragments of foreign or unusual proteins inside the cell, and display them on the surface of the cell. This allows immune cells to identify and kill the infected cells.

Cells produce many different MHC class I molecules that are able to bind to different protein fragments. Some MHC molecules can bind to a wider variety of protein fragments than others, and the number of these molecules present on the cell surface can also vary between individuals. Researchers have noticed that individuals with particular MHC molecules tend to be more or less resistant to particular diseases. For instance, individuals with certain MHC molecules tend to take longer to develop AIDS if they become infected with HIV. However, it is not clear how either the variety of protein fragments bound or the numbers of MHC class I molecules on the surface of cells could alter the immune response.

Here, Chappell, Meziane et al. studied MHC class I molecules in chickens and humans. The experiments reveal that the MHC class I molecules that can bind to a larger variety of protein fragments (so-called 'generalists') are present in lower numbers on the surface of cells than molecules that can bind to a smaller variety of fragments (so-called 'specialists').

Furthermore, generalist MHC molecules were found to provide resistance to Marek's disease in chickens—which causes paralysis—but some specialists slowed the progression of HIV infections into AIDS in humans. Chappell, Meziane et al. propose that these two types of MHC class I molecules evolved to perform different roles in immune responses. This is a new way of looking at the role of MHC molecules in fighting disease, and the next challenge is to explore the implications for medicine and evolutionary biology.

Recently, the level of cell surface expression of HLA-C alleles correlated with CD8 T cell cytotoxicity has been proposed as one important basis for control (*Thomas et al., 2009*; *Apps et al., 2013*). Understanding how cell expression level might impact on disease resistance is complicated in humans due to the presence of three class I loci, so we began by studying a simpler animal system before examining human class I alleles.

Long ago, we reported that the relative expression level of MHC class I molecules on the surface of chicken red blood cells as assessed by flow cytometry varies significantly, with cells of the MHC haplotype B21 approximately ten-fold lower than B4, B12, B15, and B19 (*Kaufman et al., 1995*). This finding was of interest because the level of cell surface expression is inversely correlated with the reported levels of MHC-determined resistance to Marek's disease, an economically important disease caused by the oncogenic herpesvirus, Marek's disease virus (MDV). Decades of investigation identified B21 (and other haplotypes like B2, B6, and B14) as generally conferring resistance and B19 (and other haplotypes such as B4, B12, and B15) as generally conferring susceptibility (reviewed in *Plachy et al., 1992*). On this basis, we proposed that MHC-determined resistance to Marek's disease could be due to the cell surface expression polymorphism of class I molecules (*Kaufman et al., 1995*; *Kaufman and Salomonsen, 1997*).

Compared to humans, the chicken MHC is relatively simple (*Kaufman et al., 1999*), with two classical class I genes BF1 and BF2 that flank the genes for the transporter associated with antigen presentation (TAP). The TAP transporter pumps peptides from the cytoplasm to the lumen of the endoplasmic reticulum for loading nascent class I molecules, and in typical mammals is functionally monomorphic, pumping a wide variety of peptides for all members of the polymorphic class I multigene family. In contrast, chicken TAP genes are highly polymorphic, with each haplotype encoding a TAP molecule with a peptide-translocation specificity matching the peptide-binding specificity of the dominantly expressed class I molecule encoded by the BF2 gene, with the BF1 gene expressed poorly or not at all (*Wallny et al., 2006*; *Shaw et al., 2007*; *Walker et al., 2011*).

In chickens, the peptide-binding specificity of the dominantly expressed class I molecule can determine resistance to infectious pathogens as well as responses to vaccines. The peptide motifs

from B4, B12, B15, and B19 class I molecules are if anything more fastidious than human and mouse motifs, with only one or two amino acids found at the positions of anchor residues, with the motifs explaining the immune response to infection and vaccination (*Kaufman et al., 1995*). The fastidious binding of these molecules could be easily understood from wire models of the binding site, in which charged and hydrophobic residues were found in appropriate places to interact in a simple manner with the anchor residues of the bound peptides (*Wallny et al., 2006*). This view was confirmed from the crystal structure of the dominantly expressed class I molecule BF2*0401 from the B4 haplotype, with positive-charged residues in a narrow groove allowing only certain anchor residues from the peptide to be accommodated (*Zhang et al., 2012*). In contrast, much less peptide material was isolated from B21 cells, with many amino acids found in every peptide position. Crystal structures demonstrated that the dominantly expressed class I molecule BF2*2101 remodels the binding site to accommodate two peptides with completely different sequences, including the anchor residues (*Koch et al., 2007*). Thus, the lack of a clear peptide motif could be explained as promiscuous peptide binding due to the remodelling of the peptide-binding site.

In this paper, we find that two other haplotypes known to confer resistance to Marek's disease also have low cell surface expression and promiscuous peptide motifs, and examine the structural basis for promiscuous binding in three low expressing molecules. We show that the same relationship between cell surface expression and peptide-binding repertoire is found for four human class I molecules, associated with progression to AIDS. Finally, we propose how promiscuous peptide binding might confer resistance to some pathogens, what the basis for the cell surface expression polymorphism might be, and how these properties may relate to different strategies for resistance to pathogens.

## Results

### Chicken class I molecules show an inverse correlation between cell surface expression and peptide-binding repertoire

We reported that the relative level of class I molecules on the surface of erythrocytes is ten-fold lower for adult chickens of the B21 MHC haplotype compared to the B4, B12, B15, and B19 haplotypes, as assessed by flow cytometry with two different monoclonal antibodies (mAb) (*Kaufman et al., 1995*). Extending this work, we examined another cell type, spleen lymphocytes, with a quantitative flow cytometric assay using a mAb to chicken heavy chain. We found a similar rank order for these and other haplotypes (*Figure 1*), with the difference between the highest and lowest ranging up to fourfold over many experiments. The B2, B14, and B21 haplotypes can be considered as 'low-expressing haplotypes', while B4, B12, B15, and B19 haplotypes are 'high-expressing haplotypes', with low-expression correlating historically with resistance to Marek's disease.

For the high-expressing haplotypes B4, B12, B15, and B19, the dominantly expressed class I molecules all have definite peptide motifs, with two or three peptide positions each bearing one or two chemically similar residues (*Kaufman et al., 1995*; *Wallny et al., 2006*). In contrast, much less peptide material was isolated from low expressing B21 cells, with pool sequences showing many amino acids in every peptide position (*Koch et al., 2007*). Extending this work, we sequenced pools of peptides from erythrocytes and leukocytes of chickens with B2 and B14 haplotypes and found that they also showed many amino acids in each position, mostly with very different chemical characteristics (*Figure 2A*). Sequencing of single peptides from B2, B14, and B21 haplotypes confirmed this

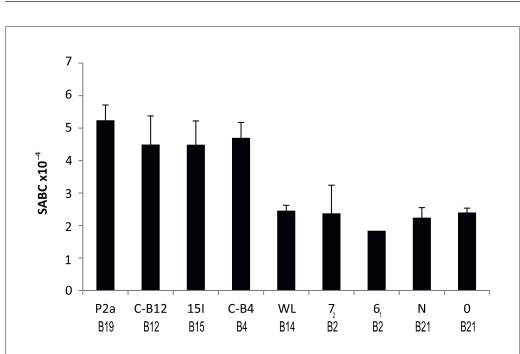

**Figure 1**. Cell surface expression levels of class I molecules vary markedly between chicken haplotypes, as determined by a quantitative flow cytometric assay. Spleen cells from various inbred experimental chicken lines (with MHC haplotypes indicated) were stained with the monoclonal antibody F21-2 against chicken major histocompatibility complex (MHC) class I heavy chain and the specific antigen binding capacity (SABC, which reflects number of epitopes per cell calculated in reference to specific antibody-binding calibration beads). Results are means of triplicate stains, with error bars indicating standard deviation.

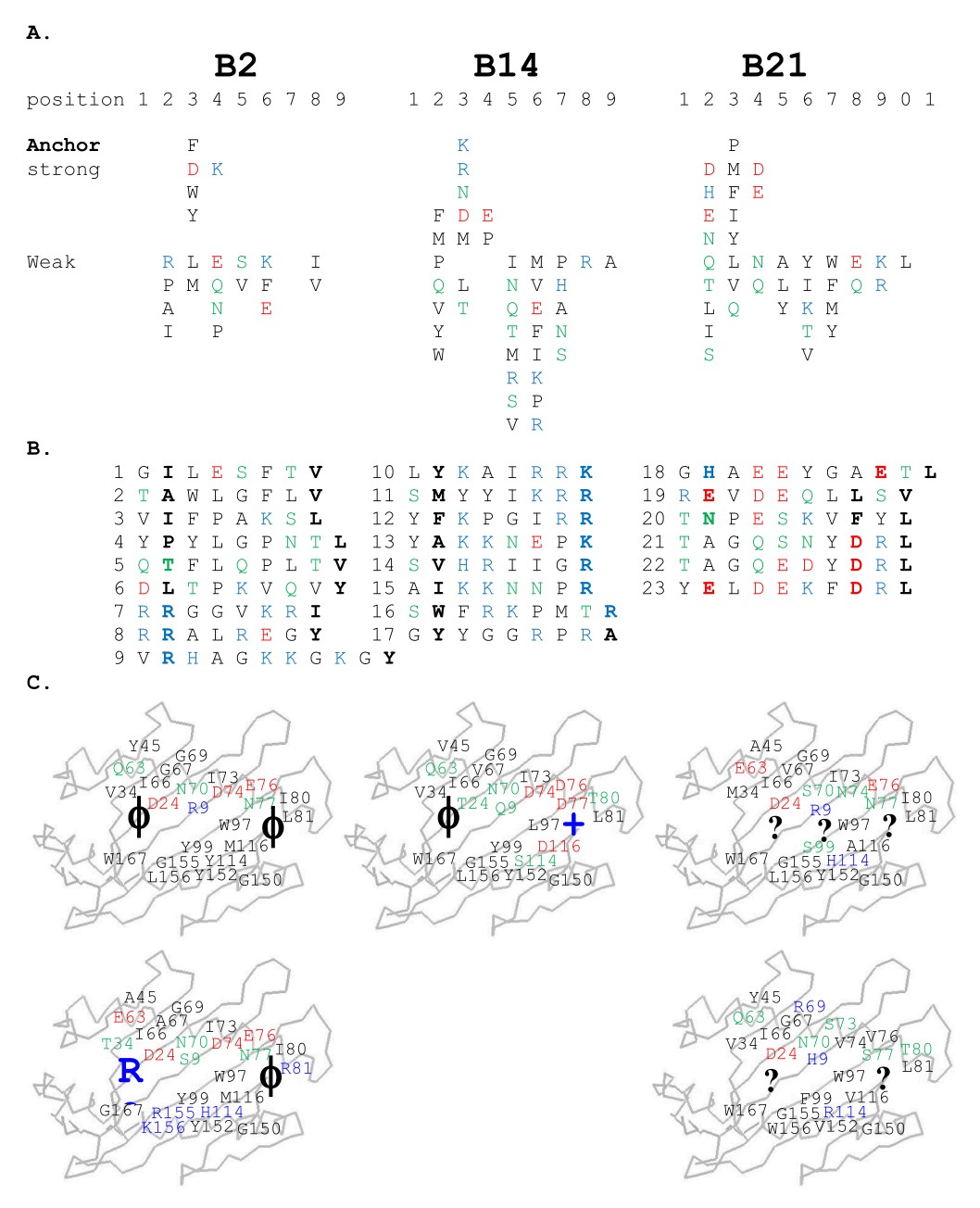

**Figure 2**. Peptides isolated from class I molecules of B2, B14, and B21 chickens show promiscuity of peptide binding. For all panels, amino acids are in single letter code, with basic residues shown in blue, acidic in red, polar in green, hydrophobic in black. (**A**) Sequences of peptides bound to class I molecules isolated from three chicken strains determined from peptide pools showing apparent anchor, strong and weak signals. (**B**) Sequences of individual peptides, with confirmed anchor residues in bold. (**C**) Peptide anchor residues in large letters (or question marks for unknown) superimposed on a model of class I α1 and α2 domains with those residues of the major (above) and minor (below) class I sequences that are both polymorphic and potentially peptide contacts indicated as smaller letters; numbering based on human class I (HLA-A2) sequence.

sequence diversity, with no obvious anchor positions bearing one or two residues with similar chemical characteristics (*Figure 2B*).

Thus, the chicken class I molecules from high-expressing haplotypes have clear peptide motifs, at least as fastidious as such motifs described in humans and mice. In contrast, the class I molecules from

low-expressing haplotypes have no obvious peptide motifs, with promiscuous binding unlike what has been described in mammals. To confirm that these properties result in different numbers of distinct peptides on the cell surface, we isolated class I molecules from equal numbers of cells from the B19 cell line 265L and the B21 cell line AVOL-1, and analysed the bound peptides by mass spectrometry. Despite the B19 line having twice as many class I molecules on the cell surface as the B21 line as assessed by quantitative flow cytometry, there were only one third as many distinct peptides identified by mass spectrometry (*Figure 3*). Thus, a promiscuous class I molecule can and does bind a greater variety of peptides that appear on the surface of the cells, in comparison to a fastidious molecule.

## The low-expressing BF2*2101 molecule remodels the binding site to accommodate a variety of peptides with very different anchor residues

Unlike high-expressing haplotypes (*Wallny et al., 2006*), the wire models of the dominantly expressed class I of the B21 haplotype, BF2*2101, gave no clue as to which peptides might bind (*Figure 2C*). Crystal structures, based on heavy chain and $\beta_2$-microglobulin ($\beta_2$m) expressed in bacteria and refolded with synthetic peptides, showed that BF2*2101 remodels the binding site to accommodate two peptides with completely different sequences, including the anchor residues (*Koch et al., 2007*). Small residues bordering the groove lead to a large cavity in the middle of the groove, within which Asp24 and Arg9 can move, creating different configurations, as illustrated for the 11mer peptide GHAEEYGAETL and the 10mer peptide REVDEQLLSV (*Figure 4A,B*). The peptide position $P_2$ His of the 11mer interacts with the Asp24, while the $P_{c-2}$ Glu interacts with the Arg9 (and $P_c$ Leu fits in a hydrophobic pocket at the end of the groove). The 10mer remodels the binding site, so that the $P_2$ Glu interacts with Asp24 that also interacts with Arg9 in a so-called charge transfer mechanism, creating a hydrophobic pocket that accommodates $P_{c-2}$ Leu (and $P_c$ Val fits in the hydrophobic pocket).

Additional peptides eluted from MHC molecules of B21 cells (*Figure 2B*) show that other amino acids at $P_2$ and $P_{c-2}$ can be accommodated by the critical Asp24 and Arg9 within the large central cavity in the binding site. In fact, the 10mer TNPESKVFYL binds BF2*2101 in similar way to the 10mer REVDEQLLSV (*Table 1*, *Figure 4C*), with $P_2$ Asn binding the Asp24 aided by charge transfer with Arg9, and with the rearrangement of Arg9 permitting the accommodation of $P_{c-2}$ Phe, just as for $P_{c-2}$ Leu in the previous structure.

However, the two structures with the 10mers TAGQEDYDRL and TAGQSNYDRL display a completely different mode of binding (*Table 1*, *Figure 4D,E*), for which the $P_2$ Ala does not interact with the MHC molecule at all (and therefore is not an anchor residue), with Arg9 interacting as a bridge between Asp24 and $P_{c-2}$ Asp. A further mode of binding is shown by the 10mer YELDEKFDRL (*Table 1*, *Figure 4F*), for which both anchor residues at positions $P_2$ and

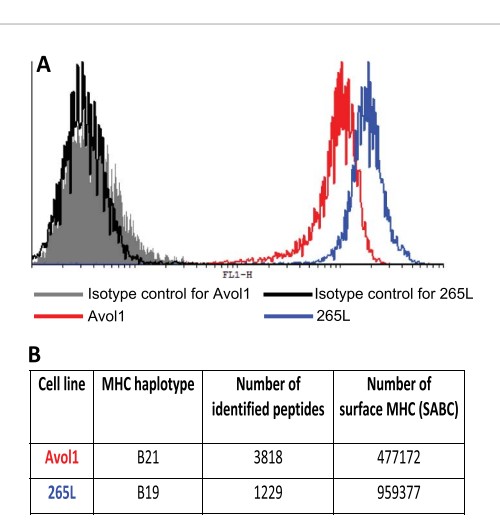

| Cell line | MHC haplotype | Number of identified peptides | Number of surface MHC (SABC) |
|---|---|---|---|
| Avol1 | B21 | 3818 | 477172 |
| 265L | B19 | 1229 | 959377 |

**Figure 3**. There is an inverse correlation between the cell surface expression levels of class I molecules and the variety of peptides isolated from class I molecules. (**A**) The B19 cell line 265L and the B21 cell line AVOL-1 were analysed by flow cytometry by staining with the mAb F21-2 to chicken class I molecules. AVOL-1 had slightly more autofluorescence, so the settings on the FACScan were adjusted so that the mean fluorescence intensity of the isotype control sample was the same as for 265L. The histogram shows the fluorescence intensity in the FL1 channel on the x-axis and the number of events on the y-axis. (**B**) In the same flow cytometry experiment, the calibration beads from the QIFIKIT were stained separately with the secondary antibody for calibration curves to calculate the SABC, which reflects the absolute numbers of epitopes on the cell surface. As a separate experiment, the class I molecules were isolated from each cell line by affinity chromatography with F21-2 and analysed by LC-MS/MS. Table shows the SABC and the number of different peptides found for each cell line.

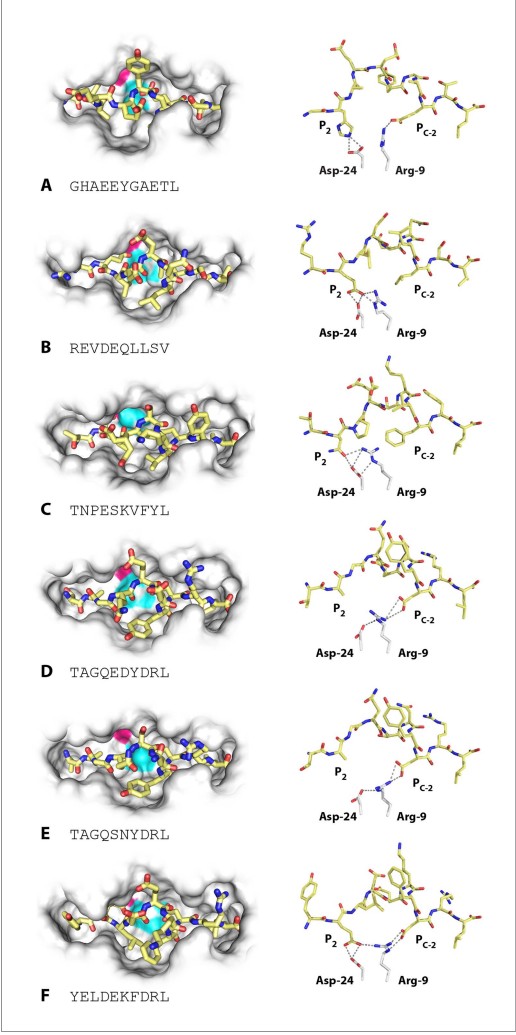

**Figure 4**. Structures of BF2*2101 with different peptides show several modes of promiscuous binding through remodelling of the binding site. Left panels, top down view with peptide as sticks (N-terminus to the left; carbon atoms, yellow; nitrogen atoms, blue; oxygen atoms, red) and class I molecule as solid surface (grey except for positions of Asp24 side chain oxygen atoms in pink and Arg9 side chain nitrogen atoms as cyan). Right panels, side view from α2 domain side with peptide, Asp24 and Arg9 as sticks (hydrogen bonds, dotted lines; carbon atoms of Asp24 and Arg9, white; all else as in left panels). (**A**) GHAEEYGAETL (peptide P316; PDB 3BEV); (**B**) REVDEQLLSV (P330; 3BEW); (**C**) TNPESKVFYL (P458; 2YEZ); (**D**) TAGQEDYDRL (P394; 4D0B); (**E**) TAGQSNYDRL (P399; 4D0C); (**F**) YELDEKFDRL (P400; 4CVZ).

$P_{c-2}$ are acidic. In this structure, the $P_2$ Glu interacts with both Asp24 and Arg9, and $P_{c-2}$ Asp interacts with Arg9. These various modes of binding are presumably just a few out of many and illustrate the promiscuous binding of BF2*2101, unlike anything seen for mammalian class I molecules.

## The low expressing BF2*0201 and BF2*1401 molecules bind a variety of peptides through broad hydrophobic pockets

Similar to B21, the peptides isolated from cells of the low expressing haplotypes B2 and B14 have no obvious motifs. Pool sequences show no position at which only one or two chemically similar amino acids are present as anchor residues (*Figure 2A*). Individual peptides have no classic pattern of anchor residues, but some features are discernable (*Figure 2B*).

The B2 peptides fall into two groups. One small group of peptides has $P_2$ Arg and mostly $P_1$ Arg and $P_c$ Tyr (*Figure 2B*), much like the dominantly expressed class I molecule of the high-expressing B15 haplotype, BF2*1501 (*Wallny et al., 2006*). These three peptides are likely to have been isolated from the poorly expressed molecule encoded by the minor gene of the B2 haplotype, BF1*0201, the wire model of which (*Figure 2C*) looks very similar to BF2*1501. Size exclusion chromatography (SEC) of heavy chain and $\beta_2$m expressed in bacteria and refolded with peptide showed that BF1*0201 but not BF2*0201 binds these peptides (*Figure 5*).

All of the other B2 peptides have generally smaller hydrophobic amino acids at $P_2$ and somewhat larger hydrophobic amino acids at $P_c$ (*Figure 2B*). Crystal structures with two of these peptides (*Table 1*, *Figure 6A,B*), YPYLGPNTL and VIFPAKSL, show that Pro and Ile at peptide position $P_2$ and Leu at $P_c$ bind shallow hydrophobic pockets (*Figure 6A,B*), similar to the way in which anchor residues bind pockets B and F in many mammalian class I molecules. This mode of binding is completely different from BF2*2101 that remodels the binding site, despite the fact that BF2*0201 shares Asp24 and Arg9 with BF2*2101. In these BF2*0201 structures, Arg9 interacts mostly with peptide main chain atoms and Asp24 interacts with Tyr43 (residue 45 being the equivalent position in human class I molecules), a residue that differs between BF2*0201 and BF2*2101, ultimately leading to a more hydrophobic pocket B to accommodate $P_2$ residues (*Figure 6A,B*).

The B14 peptides all share two features, generally a medium to large hydrophobic amino acid at $P_2$ and one or more basic amino acids at the end of the peptide (*Figure 2B*). A wire model of BF2*1401 shows small hydrophobic and polar residues in and around the beginning of the groove and reveals

**Table 1**. Data collection and refinement statistics

| PDBID | 4d0b | 4d0c | 4cvz | 2yez | 4cvx | 4d0d | 4cw1 |
|---|---|---|---|---|---|---|---|
| Data collection | | | | | | | |
| Space group | P212121 | P212121 | P212121 | P212121 | P65 | P212121 | P212121 |
| Cell dimensions | | | | | | | |
| a, b, c (Å) | 60.9 | 60.5 | 60.5 | 60.6 | 173.9 | 88.1 | 62.2 |
| | 69.2 | 68.9 | 69.0 | 69.0 | 173.9 | 92.5 | 90.58 |
| | 95.4 | 94.8 | 93.7 | 94.8 | 87.5 | 223.6 | 144.8 |
| $\alpha, \beta, \gamma$ (°) | 90 | 90 | 90 | 90 | 90 | 90 | 90 |
| | 90 | 90 | 90 | 90 | 90 | 90 | 90 |
| | 90 | 90 | 90 | 90 | 120 | 90 | 90 |
| Resolution (Å) | 56.03–2.80 (2.95–2.80) | 51.00–2.82 (2.89–2.82) | 50.84–2.39 (2.56–2.39) | 41.03–2.45 (2.58–2.45) | 75.66–3.30 (3.56–3.30) | 88.08–3.13 (3.21–3.13) | 72.38–2.58 (2.65–2.58) |
| $R_{sym}$ or $R_{merge}$ | 0.118 (0.30) | 0.12 (0.30) | 0.15 (0.91) | 0.07 (0.22) | 0.17 (0.56) | 0.18 (0.69) | 0.14 (0.59) |
| $I/\sigma I$ | 8.1 (2.8) | 8.2 (1.9) | 11.9 (2.1) | 9.5 (1.5) | 11.1 (3.8) | 8.8 (2.1) | 8.8 (2.3) |
| Completeness (%) | 91.2 (89.8) | 90.8 (76.0) | 99.9 (100) | 85.9 (53.0) | 100 (99.9) | 99.6 (98.8) | 99.3 (98.4) |
| Redundancy | 4.0 (3.6) | 3.8 (1.9) | 6.4 (6.5) | 2.8 (1.6) | 7.3 (7.5) | 4.6 (4.4) | 5.5 (4.2) |
| Refinement | | | | | | | |
| Resolution (Å) | 56.03–2.80 | 50.92–2.81 | 50.84–2.39 | 41.03–2.45 | 75.66–3.30 | 88.08–3.13 | 72.38–2.58 |
| No. reflections | 9361 | 8967 | 16,086 | 8913 | 21,707 | 32,916 | 26,228 |
| $R_{work}/R_{free}$ | 27.5/29.6% | 27.5/28.2% | 25.9/26.9% | 25.2/27.3% | 23.8/26.2% | 29.2/29.9% | 28.6/29.1% |
| Number of atoms | | | | | | | |
| Protein | 3061 | 3057 | 3079 | 3091 | 6052 | 12,102 | 6044 |
| Ligand/ion | 0 | 4 | 24 | 0 | 0 | 0 | 0 |
| Water | 5 | 31 | 42 | 30 | 8 | 9 | 77 |
| B-factors | | | | | | | |
| Protein | 30.30 | 34.24 | 30.23 | 32.22 | 77.86 | 47.02 | 40.02 |
| Ligand/ion | – | 26.32 | 46.25 | – | – | – | – |
| Water | 14.33 | 12.88 | 24.38 | 8.61 | 34.87 | 34.31 | 29.50 |
| Root mean square (r.m.s.) deviations | | | | | | | |
| Bond lengths (Å) | 0.008 | 0.007 | 0.007 | 0.007 | 0.004 | 0.007 | 0.007 |
| Bond angles (°) | 0.89 | 0.86 | 0.85 | 0.83 | 0.759 | 0.82 | 0.84 |

Highest resolution shell is shown in parenthesis.

many acidic residues at the end of the groove, well placed to interact with the basic residues at the peptide C-terminus (*Figure 2C*). The crystal structure of peptide SWFRKPMTR bound to BF2*1401 shows that Trp at position $P_2$ interacts with hydrophobic surfaces in pocket B, while Arg at $P_c$ interacts with Asp73(human position 74), Asp76(77) and Asp113(116) in pocket F (*Table 1*, *Figure 6C*). Both these pockets are much deeper in BF2*1401 compared to BF2*0201, which explains the larger anchor residues found in the B14 peptides compared to most B2 peptides. The differences in pocket sizes are primarily due to difference in size between Val43(45) and Tyr43(45) for pocket B and between Asp113 (116) and Met113(116) for pocket F, which are supported by a network of hydrogen bonds from Thr24 and Gln9 in BF2*1401 and from Asp24 and Arg9 in BF2*0201 (*Figure 6*).

These data show that the three class I molecules with lower expression on the cell surface all bind a wide range of peptides with no obvious peptide motif, although the mechanism by which the promiscuous binding is achieved is different for each molecule. One feature that may be important is the width of the peptide-binding groove (*Figure 7*), which is the narrowest in the high expressing BF2*0401 molecule (*Zhang et al., 2012*) and the widest in the low expressing BF2*2101.

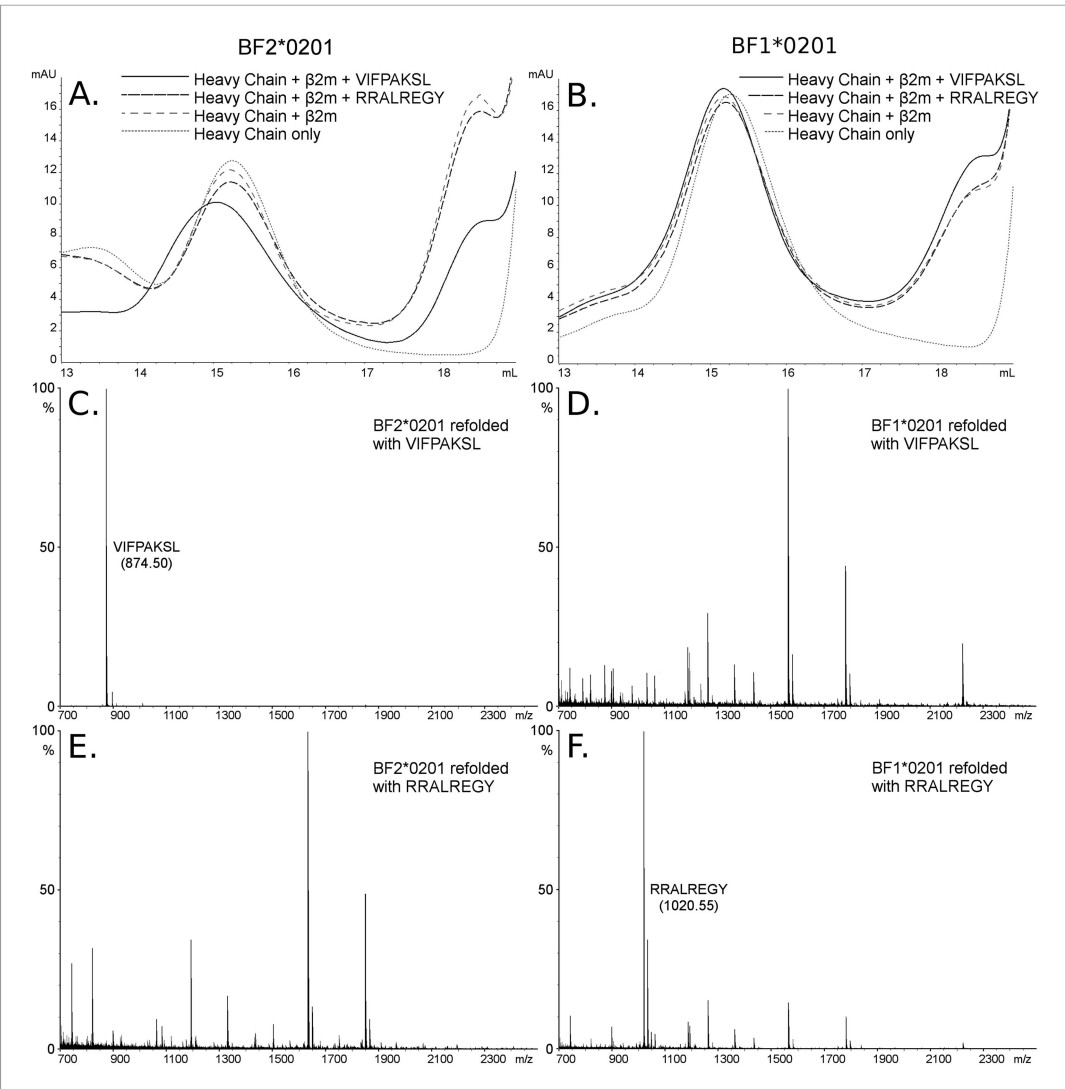

**Figure 5**. The dominantly expressed class I molecule BF2*0201 binds VIFPAKSL but not RRALREGY, while the minor class I molecule BF1*0201 binds RRALREGY but not VIFPAKSL. (**A** and **B**) Size exclusion chromatography (SEC) traces for BF2*0201 or BF1*0201 heavy chains expressed in bacteria refolded with or without $\beta_2$-microglobulin ($\beta_2$m) and peptide. The heavy chain BF2*0201 refolded with $\beta_2$m and the appropriate peptide migrates as a native monomer, whereas refolded with the inappropriate or no peptide migrates in the same position as heavy chain alone. In contrast, all these conditions for the heavy chain BF1*0201 result in molecules that migrate roughly the same mobility. (**C** through **F**) Mass spectrometry (MALDI-TOF) analysis of the monomer peaks of heavy chain refolded with $\beta_2$m and peptide shows that VIFPAKSL but not RRALREGY can be recovered from BF2*0201, while RRALREGY but not VIFPAKSL can be recovered from BF1*0201. Note the many peaks for BF1*0201 with VIFPAKSL and for BF2*0201 with RRALREGY, representing background contaminants detected as sensitivity was increased in the search for the synthetic peptide. Comparable results were obtained with YPYLGPNTL, RRALREGY, RRGGVKRI, and the B15 (BF2*1501) peptide KRLIGKRY.

## Human class I molecules also show an inverse correlation between cell surface expression and peptide-binding repertoire

The structural analysis of BF2*2101 showed that this low expressing class I molecule achieves promiscuous binding by remodelling the peptide binding site in a way never described for a mammalian class I molecule. However, analysis of BF2*0201 and BF2*1401 shows promiscuous binding by chicken molecules using anchor positions into pockets B and F, much like many mammalian classical class I molecules. In order to determine whether the inverse correlation between cell surface

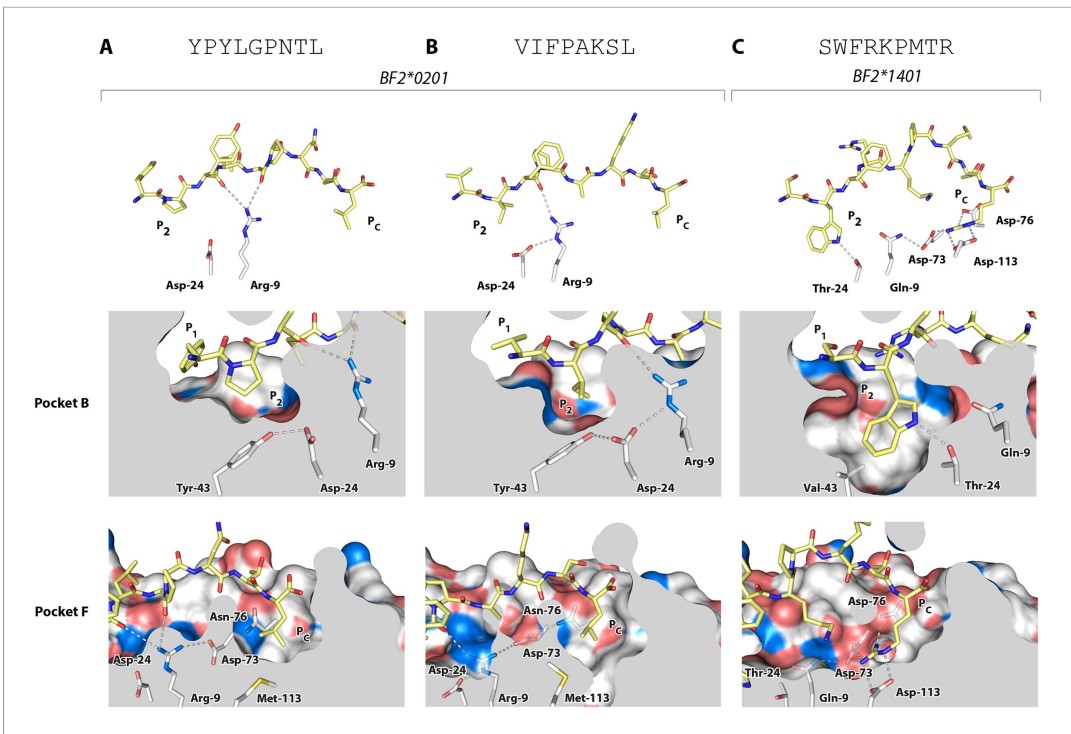

**Figure 6**. Structures of BF2*0201 and BF2*1401 show promiscuous binding via hydrophobic binding pockets for the anchor residues at peptide position $P_2$ and $P_c$, with the class I residues at positions 24 and 9 playing supporting roles, and with residues lining the pockets explaining the relative size of anchor residues. Upper panels, side view from $\alpha2$ domain side with peptide as sticks (N-terminus of peptide to the left; carbon atoms of peptide, yellow; carbon atoms of class I molecule, white; nitrogen atoms, blue; oxygen atoms, red; hydrogen bonds, dotted lines; carbon atoms of Asp24 and Arg9, white). Middle (pocket B) and bottom (pocket F) panels, side view cut-away from $\alpha2$ domain side with peptide and selected class I residues as sticks (numbering based on chicken class I sequence) and with rest of MHC molecule as solid surface. (**A**) YPYLGPNTL bound to BF2*0201 (peptide P377; PDB 4CVX); (**B**) VIFPAKSL bound to BF2*0201 (P473; 4D0D); (**C**) SWFRKPMTR bound to BF2*1401 (P479; 4CW1).

expression level and peptide binding promiscuity is a feature just of chickens or is indicative of a more fundamental property, we examined some human class I alleles.

Only a few studies have explored the extent of the peptide-binding repertoire of different human class I molecules. One of these studies reported the predicted peptide-binding repertoires for four human class I alleles, finding a rank hierarchy from the extremely fastidious HLA-B*57:01 to HLA-B*27:05 to HLA-B*07:02 to the highly promiscuous HLA-B*35:01, that correlated directly with progression from HIV infection to AIDS (*Kosmrlj et al., 2010*). The rank hierarchy of these alleles is the same as that determined by binding of peptide libraries (*Paul et al., 2013*).

We identified two mAbs reported (*Apps et al., 2009*) to react with all four HLA-B alleles (along with HLA-C alleles, which are poorly expressed on blood cells), but not with certain HLA-A alleles. Volunteers with the proper combinations of homozygous HLA-A and HLA-B alleles were recruited from a large group of bone marrow donors (*Table 2*), and the cell surface expression levels on their blood lymphocytes and monocytes were examined by quantitative flow cytometry. The two mAb (Tu149 and B1.23.2) reacted with all four HLA-B alleles and gave similar (but not identical) results for both lymphocytes and monocytes, with a rank hierarchy in cell surface expression ranging from HLA-B*57:01 as the highest to HLA-B*27:05 to HLA-B*07:02 to HLA-B*35:01 as the lowest (*Figure 8A–D*).

One potential concern with these data was that the difference in expression level of the HLA-B alleles might actually be due to differences in HLA-C levels, which are known to vary between alleles (*Thomas et al., 2009*; *Apps et al., 2013*). However, inspection shows that there is no correlation of expression levels reported for the HLA-C alleles found for the individual donors and the expression level of the mAb binding (*Table 2*), if anything the reverse with the low expressing HLA-B*35:01 donors all having HLA-C alleles reported to have high expression.

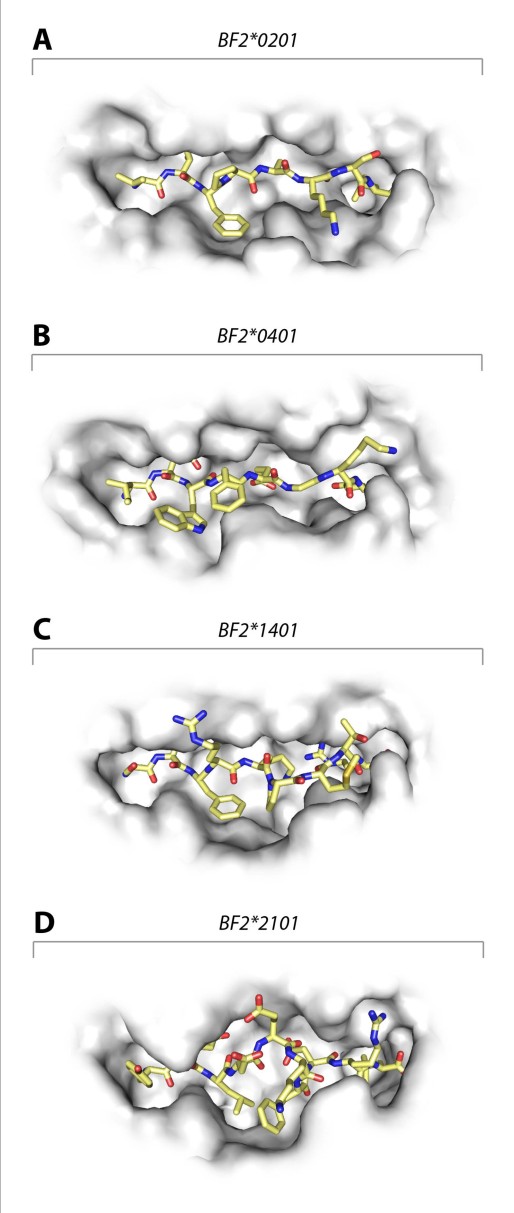

**Figure 7**. Structures of chicken class I molecules show differences in the width of the peptide-binding groove, with the fastidious BF2*0401 having the narrowest groove and the promiscuous BF2*2101 being the widest in the centre of the groove. Top down view with peptide as sticks (N-terminus to the left; carbon atoms, yellow; nitrogen atoms, blue; oxygen atoms, red) and class I molecule as grey solid surface. (**A**) VIFPAKSL bound to BF2*0201 (P473; 4D0D); (**B**) IDWFEGKE bound to BF2*0401 (IE8; 4G43); (**C**) SWFRKPMTR bound to BF2*1401 (P479; 4CW1); (**D**) YELDEKFDRL bound to BF2*2101 (P400; 4CVZ).

Another concern was that the difference in expression level of the HLA-B alleles actually reflects differences in affinity of binding by the two mAb. The fact that both antibodies gave similar but non-identical results suggests that they do not recognize exactly the same epitopes and therefore are unlikely by chance to vary similarly in affinity. We tested whether the two antibodies bound independently and found that B1.23.2 inhibited both itself and Tu149 very well, while Tu149 inhibited both itself and B1.23.2 much less well (*Figure 9*). These results suggest that B1.23.2 has a much higher affinity than Tu149 and that their epitopes overlap. We also utilized a third antibody (22E-1) that reacts with the HLA-B*57:01 and HLA-B*27:05 but not HLA-B*07:02 and HLA-B*35:01 (nor with any HLA-C allele), and therefore must recognize a different epitope. We found the same relationship of expression level for HLA-B*57:01 and HLA-B*27:05 as with the other two antibodies (*Figure 8E,F*).

Thus, in humans the high expressing class I molecules have fastidious peptide binding, and the low expressing molecules have promiscuous binding. We conclude that the correlation between peptide-binding repertoire and expression level is the same in chickens and humans and is indicative of a fundamental property of class I molecules.

## Discussion

In this article, we make three major points. First, we extend previous work showing that the chicken BF2*2101 achieves promiscuity by remodelling the peptide binding site in a way unknown for mammals, but we find that two other low expressing chicken class I molecules achieve promiscuity with the same pockets typically used in mammals. By extension, these latter results provide a potential molecular explanation for recent reports that human class I molecules vary in peptide repertoire and other reports that they vary in cell surface expression level. Second, we show that peptide-binding repertoire and level of cell surface expression are inversely correlated in both chickens and humans, indicating that this relationship is a fundamental property of class I molecules. Finally, we find that there is a clear association with resistance to infectious diseases, with the low expressing promiscuous class I molecules associated with resistance to Marek's disease in chickens but associated with faster progression to AIDS in humans. We discuss each of these points in turn.

The first point is the structural and mechanistic basis for differences in peptide repertoire of chicken class I molecules. In this article, we confirm and extend the finding (*Koch et al., 2007*) that

**Table 2**. Anonymized donors with class I alleles and inferred HLA-C expression

| Donor | HLA-A alleles | HLA-B alleles | HLA-C alleles | HLA-C expression |
|-------|---------------|---------------|---------------|------------------|
| 5701/02 | 02:01, 02:01 | 57:01, 57:01 | 06:02, 15:02 | high, unknown |
| 5701/03 | 01:01, 01:01 | 57:01, 57:01 | 06:02, 06:02 | high, high |
| 5701/04 | 01:01, 03:01 | 57:01, 57:01 | 06:02, 07:01 | high, low |
| 2705/1 | 03:01, 68:01 | 27:05, 27:05 | 01:02, 02:02 | high, high |
| 2705/2 | 02:06, 11:01 | 27:05, 27:05 | 01:02, 03:03 | high, low |
| 2705/3 | 02:01, 02:01 | 27:05, 27:05 | 01:02, 12:03 | high, high |
| 2705/4 | 02:01, 02:12 | 27:05, 27:05 | unknown | |
| 0702/1 | 02:01, 03:01 | 07:02, 07:02 | 07:02, 07:02 | low, low |
| 0702/2 | 03:01, 03:01 | 07:02, 07:02 | 07:02, 07:02 | low, low |
| 0702/3 | 02:01, 03:01 | 07:02, 07:02 | 07:02, 07:02 | low, low |
| 0702/4 | 03:01, 11:01 | 07:02, 07:02 | 07:02, 07:02 | low, low |
| 3501/1 | 11:01, 11:01 | 35:01, 35:01 | 04:01, 04:01 | high, high |
| 3501/2 | 11:01, 11:01 | 35:01, 35:01 | 04:01, 04:01 | high, high |
| 3501/3 | 03:01, 03:01 | 35:01, 35:01 | 04:01, 04:01 | high, high |
| 3501/4 | 11:01, 11:01 | 35:01, 35:01 | 04:01, 04:01 | high, high |

BF2*2101 binds 10mer and 11mer peptides with three anchor positions, with many different amino acids with different chemical characteristics, mostly charged and polar at positions $P_2$ and $P_{c-2}$ (and hydrophobic at $P_c$), which is unlike any mammalian molecule described thus far. In contrast, we find that BF2*0201 and BF2*1401 bind 8mer and 9mer peptides with two anchor residues (much like typical mammalian class I molecules), each with a variety of amino acids at $P_2$ and $P_c$.

One question is why BF2*2101 and BF2*0201 bind in such different ways, given that both have Asp24 and Arg9 in their binding sites. In part, this may be due to the difference in residues lining the peptide-binding groove. In BF2*2101, small residues like Gly68(human position 69), Ser69 (70), Ser97(99), and Gly152(155) create a big cavity in the centre of the binding groove, allowing Asp24 and Arg9 to move. In BF2*0201 and BF2*1401, the two Ser residues are replaced by the large residues Asn69(70) and Tyr97(99), creating a constricted groove much more like narrow grooves found in high expressing fastidious molecules such as BF2*0401. It is also possible that the length of peptides is determined at least in part by TAP specificity. Together, the narrower groove and the shorter peptides ensure that Asp24 and Arg9 in BF2*0201 (as well as Thr24 and Gln9 in BF2*1401) cannot reach the $P_2$ and $P_{c-2}$ positions, and thus they interact (if at all) with the main chain atoms of the peptide.

Another question is how chicken class I molecules BF2*0201 and BF2*1401 compare to human molecules like HLA-A2, all of which bind

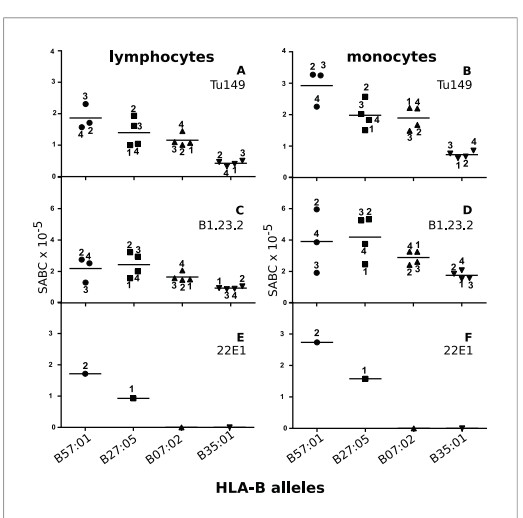

**Figure 8**. Human class I molecules show an inverse correlation between cell surface expression level and peptide binding promiscuity. Levels of specific antibody binding capacity (SABC) for different mAb: (**A** and **B**) Tu149, (**C** and **D**) B1.2.23, (**E** and **F**) 22E-1 for (**A**, **C** and **E**) ex vivo lymphoctyes and (**B**, **D** and **F**) ex vivo monocytes. Each point represents the sample from a particular donor (identified with anonymous labels that correlate with haplotypes in **Table 2**; donor B57:01/01 failed to donate); bars indicate the mean for each HLA-B allele. DOI: 10.7554/eLife.05345.012

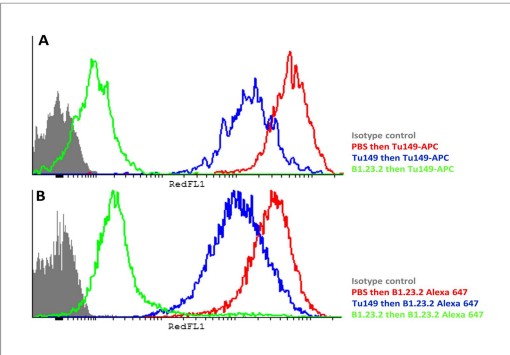

**Figure 9**. The epitopes on HLA-B57:01 for mAb B1.23.2 and Tu149 are overlapping, with B1.23.2 having a much higher affinity. Win cells were treated with saturating amounts of unlabelled Tu149 (blue), saturating amounts of unlabelled B1.23.2 (green) or PBS (red), and then stained with either **A** Tu149 conjugated to APC or **B** B1.23.2 conjugated to Alexa Fluor 647. Results were compared to staining with only isotype control (grey).

peptides with hydrophobic anchor residues. HLA-A2 has one of the widest peptide repertoires among human class I molecules, as assessed by peptide-binding studies (*Paul et al., 2013*). The anchor residues for HLA-A2 fit into narrow and specific-binding pockets, which allow almost only $P_2$ Met and Leu for pocket B and $P_c$ Val and Leu for pocket F (*Guo et al., 1993*). Although Leu and Val in particular are relatively common in proteins (so that the number of peptides bearing these amino acids should be relatively high), there is a much larger number of hydrophobic amino acids that are accommodated by the shallower pockets in both BF2*0201 and BF2*1401. Hence, it seems likely that the peptide repertoire of the most promiscuous chicken class I molecules will be greater than the most promiscuous human class I molecules.

Finally, are there obvious mechanisms by which differences in peptide repertoire arise? In chickens, the peptides presented depend on the peptide-binding specificities of the class I molecule, but also on the peptide-translocation specificities of the TAPs and perhaps also on peptide-editing by tapasin. The genes for chicken TAP and tapasin have at least as many alleles as class I genes (*Walker et al., 2005*, *2011*; *van Hateren et al., 2013*). On this basis, we propose that the difference in peptide repertoire (and in cell surface expression level) between chicken class I alleles is determined at the level of peptide loading and editing, likely to take place mostly in the peptide-loading complex. It might not be immediately apparent whether such an explanation could account for the differences in peptide repertoire and cell surface expression of human class I molecules, since human TAPs and tapasin are functionally monomorphic. However, a recent paper suggests that the assembly and cell surface expression of different HLA-B alleles is determined by interaction with the near invariant tapasin molecule, based on variation in the certain HLA-B residues (*Rizvi et al., 2014*). Indeed, it is striking that the rank order of expression level that we find for the four HLA-B alleles tested is exactly the same as the rank order of tapasin dependence found in this recent paper.

The second important point for discussion is why the cell surface expression would be inversely correlated with peptide-binding repertoire. A proximal explanation might be that the biochemical mechanism of peptide loading, as mentioned in the paragraph above, leads to this relationship. Another explanation, similar to the suggestion that multiple MHC molecules lead to a greater chance of autoimmunity (*Kaufman et al., 1995*), is that promiscuous class I molecules confer more resistance to pathogens but also lead to greater autoimmunity compared to fastidious class I molecules.

We favour a third possibility, prompted by the experimental and supporting theoretical evidence for the notion that a greater number of MHC loci would lead to greater level of negative selection in the thymus (*Vidović and Matzinger, 1988*; *Nowak et al., 1992*). In a similar way, one might expect that class I molecules presenting a lesser variety of peptides would negatively select fewer T cell clones, but class I molecules presenting a greater variety of peptides would negatively select many more T cells and create a pauperized T cell repertoire. Therefore, we propose that the peptide-binding repertoire and the cell surface expression levels are inversely correlated precisely to allow similar numbers of T cell clones to survive negative selection and be available in the periphery. In this view, low expressing and promiscuous class I molecules might present many more peptides but each individual peptide would be present on the surface at a much lower level, so that T cells would escape the negative selection that might have occurred at a higher concentration of a particular MHC-peptide complex. In essence, we propose that peptide-binding repertoires and cell surface expression levels have evolved to optimize peripheral T cell responses. Many papers have examined naïve CD8 T cell repertoire in humans and mice (for example, *Alanio et al., 2010*; *Jenkins and Moon, 2012*; *Lo et al., 2014*); such analyses with high and low expressing haplotypes would be informative.

The third important point for discussion is the fact that the inverse correlation between peptide binding repertoire and cell surface expression level also correlates with resistance and susceptibility to certain infectious pathogens. One important question is how it might work.

In chickens, these phenomena are correlated with MHC-determined resistance to historic Marek's disease. Many genetic loci contribute to resistance to Marek's disease, but historically the B locus (containing the MHC) was by far the most important, and under strong selection by MDV (*Plachy et al., 1992*; *Vallejo et al., 1998*). Here, we show that the low expressing promiscuous class I molecules are associated with resistance to Marek's disease, while the high expressing fastidious class I molecules are associated with susceptibility. A consequence of promiscuous binding is that these low expressing class I molecules present a larger repertoire of peptides which should activate a wider range of T cell clones, which may be beneficial in the immune response to certain pathogens. We propose that this breadth of antigen presentation leads to a breadth of T cell response that is the basis of MHC-determined resistance to Marek's disease. Conversely, a narrow T cell response to Marek's disease would be the basis for MHC-determined susceptibility, and there is already evidence for a limited repertoire of CD8 T cell clones infiltrating tumours in susceptible B19 chickens (*Mwangi et al., 2011*). Parenthetically, if true this model suggests that the narrowing of CD8 T cell clonality characteristic of responses in humans and mice (*Yewdell and Bennink, 1999*; *Yewdell and Del Val, 2004*; *Akram and Inman, 2012*) may not be a feature of responses involving promiscuous class I molecules in chickens.

For our example in humans, however, we find the opposite disease association. We find that the low expressing promiscuous HLA-B alleles are associated with rapid progression from HIV infection to AIDS, while the high expressing fastidious HLA-B alleles are found in non-progressors (*Carrington et al., 1999*; *Goulder and Walker, 2012*; *International HIV Controllers Study et al., 2013*). The correlation with peptide binding repertoire (as assessed by prediction) of HLA-B alleles had already been noted (*Kosmrlj et al., 2010*), but the relationship with cell surface expression level has not. The controller alleles HLA-B*57:01 and HLA-B*27:05 are known to bind and present particular viral peptides that are both protective and difficult for the virus to change without a loss in fitness (*Gillespie et al., 2006*; *Schneidewind et al., 2007*). Thus, particular class I molecules may be selected because their fastidious motif presents a particularly protective peptide.

Is there an obvious evolutionary basis for differences in peptide-binding repertoire? The fact that the fastidious HLA-B*57:01 confers resistance to AIDS while the promiscuous BF2*2101 confers resistance to Marek's disease may be no paradox, in that breadth of peptide presentation may be an appropriate response to some pathogens, while presentation of a particular peptide may be more suitable for other pathogens. Viewed from this perspective, the work presented in this paper has begun to define two groups (or a range between two extremes) of class I molecules that are strategically evolved for different modes of resistance.

One possible strategy is that fastidious class I molecules with pauci-clonal T cell responses might be better suited for rapidly evolving viruses with a limited scope for immune evasion, for which a particular peptide might be the most efficient way to achieve resistance. In contrast, promiscuous class I molecules might be better suited for pathogens with coding potential for many immune evasion genes and more stability over evolutionary time, such as large DNA viruses, bacteria, and even parasites. There are few reports about pathogen peptides that are presented in chickens, but the large literature on the MHC restriction of viruses, bacteria, and parasites in humans and mice (for example, *Fitzmaurice et al., 2015*; *Goulder and Walker, 2012*; *Macnamara et al., 2010*; *Moss and Khan, 2004*; *Steven et al., 1997*; *Yim and Selvaraj, 2010*) may be reinterpreted, as the peptide repertoire and cell surface expression levels of the class I molecules become known.

Another potentially fruitful way of considering the strategies of class I molecules (not necessarily exclusive of the first view) might be as generalists and specialists. There is a large literature in biology that examines the generalist/specialist paradigm at the species level, typically testing the model that 'the jack of all trades is the master of none' (*McArthur, 1972*; *Futuyma and Moreno, 1988*). Recent examples, among many others, include breadth of diet by insect herbivore species (*Ali and Agrawal, 2012*), niche breadth by bird species (*Julliard et al., 2006*), competition on shared hosts by aphid parasitoids (*Straub et al., 2011*), and weasel predation of voles (*Sundell and Ylönen, 2008*), as well as mathematical models to disentangle the contributions by various factors (*Remold, 2012*; *Büchi and Vuilleumier, 2014*). Indeed, the terms generalist and specialist MHC alleles have already been used for describing correlations of class II alleles of the striped mouse in Africa with number of nematode species carried (*Froeschke and Sommer, 2012*).

In this view, the promiscuous class I molecules would be generalists, which one might expect to suffice for protection against a wide variety of the most common pathogens. However, such generalists might not suffice for protection from a new and/or an especially virulent pathogen that suddenly appears, at which point there would be a strong selection for a specialist class I molecule that was particularly suited to deal with the new threat. The properties of fastidious class I molecules are consistent with selection as specialists for particular pathogens, perhaps including those no longer a danger in the current population.

The importance of these concepts for immunology and medicine may be clear from the discussion above, but there are also ramifications for evolutionary biology, ecology, and conservation. As one example, the number of MHC alleles is considered a key measure for population diversity in estimating the risk of extinction, both as a measure of overall genome diversity and in terms of fitness, including resilience to infection (*Sommer, 2005*). However, a population with a few generalist MHC alleles might remain healthy compared to a population with many inappropriate specialist MHC alleles.

## Materials and methods

### Cells

Ex vivo chicken cells were from inbred chicken lines with known MHC haplotypes (*Shaw et al., 2007*), kept at the University of Cambridge. All procedures were performed under appropriate Home Office Licenses and after review by the Ethics Committee at the University of Cambridge. Spleens were mashed through 100-μm nylon cell strainers (Falcon) in RPMI-1640 medium, supernatants taken after 5 min settling, and spleen cells recovered after centrifugation at 400×$g$ for 5 min. All procedures were carried out under Home Office licenses and with ethical approval.

Chicken cell lines were from the Pirbright Institute. AVOL-1 was derived from in vitro transformation of spleen cells from a line 0 (B21) chicken by the reticuloendotheliosis virus REV-T (*Smith, 2004*; *Yao et al., 2008*; W Mwangi and V Nair, unpublished data). The MDCC-265L cell line was established from a liver lymphoma of a line P2a (B19) chicken infected with the RB-1B virus derived from a BAC clone (as in *Yao et al., 2009*; W Mwangi and V Nair, unpublished data). Both lines were maintained in RPMI 1640 medium containing 10% foetal bovine serum, 10% tryptose phosphate broth and 1% sodium pyruvate, and at 38.5°C in 5% $CO_2$.

Ex vivo human cells were from Anthony Nolan registrants typed as homozygous for particular HLA-A and HLA-B locus alleles, who signed written consent forms, and with all procedures carried out under Human Tissue Act licenses and with ethical approval. Blood samples were collected by general practice or Walk-in Clinic phlebotomists and were couriered to the Anthony Nolan Research Institute within 24 hr. Whole blood was diluted 1:1 upon arrival with transport media (RPMI 1640 [Lonza, Belgium] supplemented with 0.6% tri-sodium citrate and 50 nM 2-mercaptoethanol), and most samples were rocked at room temperature overnight. In a similar manner as originally described for cord blood (*Figueroa-Tentori et al., 2008*), peripheral blood mononuclear cells were isolated (all steps at 20°C) using a density gradient centrifugation (Ficoll–Paque Plus 1077, GE Healthcare) at 840×$g$ for 30 min with no brake, with the buffy coat washed twice with two volumes RPMI-1640 media, spun once at 680×$g$ for 10 min and once at 540×$g$ for 10 min. All samples were stained, fixed, and anonymized before transfer to Cambridge for analysis by flow cytometry.

The human homozygous HLA-B*57 typing cell line WIN (alias IHW9095 from 10[th] International Workshop, gift of W Bultitude and S Marsh, Anthony Nolan Research Institute) was maintained in RPMI1640 with 10% foetal bovine serum and 1 mM glutamine in 5% $CO_2$ at 37°C.

### Flow cytometry

As described by instructions for quantitative flow cytometry from manufacturer (QIFIKIT, Dako) and following previous work (*Smith and Ellis, 1999*), 5 × 10[5] cells were incubated on ice in 96-well (U-bottom for chicken, V-bottom for human) microtiter plates (Nunc) with saturating primary antibody followed by washing and then by incubation with goat anti-mouse secondary antibody conjugated to fluorescein followed by washing, and data acquired using a FACscan (Becton-Dickenson). Set-up beads and calibration beads were stained separately with the secondary antibody for calibration curves to calculate the specific antigen binding capacity, which reflects the absolute numbers of

epitopes on the cell surface. Primary mAb include 200 µl tissue culture supernatant of mouse mAb F21-2 for chicken class I molecules (*Crone et al., 1985*) and three that react with certain HLA-B antigens but not certain HLA-A alleles (*Apps et al., 2009*): 200 µl tissue culture supernatant of Tu149 (*Uchańska-Ziegler et al., 1993*; gift of J Trowsdale, University of Cambridge), 20 µl 1 mg/ml B1.2.23 (*Rebaï and Malissen, 1983*; bought from eBioscience) diluted in PBS and 20 µl 1 mg/ml 22E1 (*Tahara et al., 1990*; bought from Caltag Medsystems) diluted in PBS. Saturation was confirmed by staining with dilutions of antibodies on each set of cells, except for human ex vivo cells, for which saturation was confirmed by staining WIN cells.

For the inhibition assay, Tu149 conjugated to APC (Invitrogen; kind gift of S Ashraf and J Trowsdale, University of Cambridge) and purified B1.23.2 conjugated to Alexa Fluor 647 (purified mAb from eBioscience conjugated using the Antibody Labeling kit, Molecular Probes/Life Technologies, according to manufacturer's instructions) were used. As above, $5 \times 10^5$ WIN cells were incubated with saturating amounts of unlabelled B1.23.2 or Tu149 (or with PBS) for 1 hr, washed and then incubated with a directly labelled antibody for 1 hr, before washing the cells again. Data were acquired using the red laser on a Cytek FACSanalyser (Becton Dickinson).

## Peptides from ex vivo chicken cells

Inbred chicken lines, with known MHC haplotypes (*Shaw et al., 2007*), were kept at the Institute for Animal Health at Compton. All procedures were performed under appropriate Home Office Licenses and after review by the Ethics Committee at the Institute for Animal Health. As previously described (*Wallny et al., 2006*; *Koch et al., 2007*), erythrocytes (or whole mashed up spleens) were solubilized in detergent, and class I molecules were isolated by affinity chromatography using mAb F21-2 against chicken class I molecules or F21-21 against chicken $\beta_2$m. Peptides were eluted using trifluoroacetic acid and separated by reverse phase high-pressure liquid chromatography with single peaks of abundant peptides as well as pools of non-abundant peptides sequenced by Edman degradation.

## Peptides from chicken cell lines

Immunoaffinity beads were produced, with all steps at room temperature. Protein G-Sepharose beads (Expedeon) were washed with 50 mM borate, 50 mM KCl, pH 8.0, the equivalent of 1-ml packed beads was incubated with 3 mg F21-2 (produced by the Microbiological Media Services of the Pirbright Institute) for 1 hr, treated with 40 mM dimethyl pimelimidate dihydrochloride (Sigma) in 0.1 M triethanolamine, pH 8.3 for 1 hr to cross-link the antibody to the protein G, washed with 100 mM citric acid pH 3.0, and equilibrated in 50 mM Tris, pH 8.0. The two cell lines AVOL1 and 265L were washed with PBS. Pellets of $10^9$ cells were incubated with 10 ml 1% Igepal 630, 300 mM NaCl, 100 mM Tris pH 8.0 for 30 min at 4°C, subcellular debris was pelleted by centrifugation at 300×g for 10 min and 15,000×g for 30 min at 4°C, and the cleared lysates were incubated with 1 ml immunoaffinity beads for 1 hr at 4°C. The beads were washed with 50 mM TrisCl, pH 8.0 buffer, first with 150 mM NaCl, then with 400 mM NaCl and finally with no salt. Bound material was eluted with 10% acetic acid. The eluted material was dried, resuspended in 3% acetonitrile, 0.1% formic acid in water, loaded directly onto on a 4.6 × 50 mm ProSwiftTM RP-1S column (ThermoFisher) and eluted at 500 µl/min flow rate for 10 min with a linear gradient from 2 to 35% buffer B (0.1% formic acid in acetonitrile) in buffer A (0.1% formic acid in water) using an Ultimate 3000 HPLC system (ThermoFisher), with fractions collected from 2 to 15 min. Protein detection was performed at 280 nm absorbance, with fractions eluting before $\beta_2$m pooled and dried.

For liquid chromatography tandem mass spectrometry (LC-MS/MS), peptides were analysed using either a Q-Exactive (Thermo Scientific) or a TripleTOF 5600 (AB Sciex) system. For the Q-Exactive system, peptides were separated on a Ultimate 3000 RSLCnano System utilizing a PepMap C18 column, 2 µm particle size, 75 µm × 50 cm (Thermo Scientific) with a linear gradient from 3% to 35% buffer B in buffer A (as above) at a flow rate of 250 nl/min (~65 MPa) for 60 min, and the 15 most intense precursors per full MS scan were selected for MS/MS analysis using HCD fragmentation. For the TripleTOF system, peptides were separated with a 15 cm × 75 µm ChromXP C18-CL (3 µm particle size) using an ekspert nanoLC 400 cHiPLC system (Eksigent) with a linear gradient from 8% to 35% buffer B in buffer A (as above) at a flow rate of 300 nl/min

**Table 3**. Crystallization conditions, data collection, and refinement information

| Structure (PDB ID) | Crystallization conditions | Cryo-protectant | Beamline | Data processing | Refinement program |
|---|---|---|---|---|---|
| 4 d0b | 0.1M MMT buffer, pH 5.0, 25% PEG 1500 | 15% ethylene glycol | I02 (Diamond Light Source) | AUTOPROC SUITE with XDS & SCALA | AUTOBUSTER |
| 4d0c | 0.05M KH$_2$PO$_4$, 20% PEG 8000 | 20% ethylene glycol | I02 (Diamond Light Source) | XIA2 with XDS & AIMLESS | AUTOBUSTER |
| 2yez | 0.1M MMT buffer, pH 4.0, 25% PEG 1500 | 15% ethylene glycol | I03 (Diamond Light Source) | AUTOPROC SUITE with XDS & SCALA | AUTOBUSTER |
| 4cvz | 0.1M sodium acetate, pH 5.0, 1.5 M ammonium sulphate | 8M sodium formate | I04-1 (Diamond Light Source) | XIA2 with XDS & AIMLESS | AUTOBUSTER |
| 4cvx | 0.1M MMT buffer, pH 7.0, 25% PEG 1500 | 15% ethylene glycol | I04 (Diamond Light Source) | XIA2 with XDS & AIMLESS | REFMAC |
| 4 d0d | 0.1M MMT buffer, pH 7.0, 25% PEG 1500 | 15% ethylene glycol | I04 (Diamond Light Source) | XIA2 with XDS & AIMLESS | AUTOBUSTER |
| 4cw1 | 0.1M MIB buffer, pH 5.0, 25% PEG 1500 | 15% ethylene glycol | ID29 (ESRF) | XIA2 with XDS & AIMLESS | AUTOBUSTER |

(~1600 psi) for 60 min, and CID fragmentation was induced on the 30 most abundant ions per full MS scan. All fragmented precursor ions were actively excluded from repeated selection for 15 s. Data were analysed using Peaks 7 (Bioinformatics Solutions) using a database containing all 24,092 Uniprot entries for the organism *Gallus gallus* combined with protein translations (>8 amino acids) of either all six reading frames of gallid herpesvirus 2 (NCBI entry NC_002229.3; 10,026 entries) or reticuloendotheliosis virus (NC_006934.1; 412 entries). Results were filtered using a false discovery rate of 1% that was determined by parallel searching of a randomized decoy database.

## Expression and assembly of class I molecules

As previously described (*Koch et al., 2007*), the extracellular sequence of mature BF2*2101 heavy chain and chicken $\beta_2$m cloned in pET22b(+) were expressed separately as inclusion bodies in BL21 ($\lambda$DE3) pLysS Rosetta bacterial cells, solubilized in urea and assembled together with synthetic peptide (synthesized commercially by fluorenyl-methoxy-carbonyl [fMOC] chemistry) by dilution in a renaturation buffer. Assembled class I molecules were isolated after SEC using a HiLoad 26/60 Superdex 200 column (GE Healthcare, UK). Similar procedures were used for BF1*0201, BF2*0201 and BF2*1401, except that a codon-optimized gene was synthesized (GenScript) for BF2*0201, which was then cloned into a pET28a(+) vector followed by a factor X site for cleavage, a BirA site for biotinylation and a His tag for purification (*Leisner et al., 2008*).

**Table 4**. Ramachandran statistics

| Structure (PDB ID) | Ramachandran outliers number (%) | Ramachandran favoured number (%) |
|---|---|---|
| 4cvz | 0 (0%) | 371 (98.15%) |
| 4 d0b | 0 (0%) | 356 (94.43%) |
| 4d0c | 0 (0%) | 361 (95.76%) |
| 2yez | 1 (0.26%) | 364 (96.04%) |
| 4cvx | 0 (0%) | 710 (95.69%) |
| 4 d0d | 0 (0%) | 1430 (96.30%) |
| 4cw1 | 0 (0%) | 730 (98.25%) |

## Assembly assays

As previously described (*Koch et al., 2007*), heavy chains and $\beta_2$m were expressed in bacteria as inclusion bodies and denatured in urea; $\beta_2$m was refolded and purified by SEC. For each renaturation sample, solutions containing 30 μg heavy chain plus or minus 42 μg $\beta_2$m and/or 10 μg synthetic peptide were added to refold buffer and incubated at 4°C with stirring for roughly 40 hr. After centrifugation and passage through a 0.45-μm sterile filter, each sample was loaded on a Superdex 200 10/300 GL SEC column (GE Healthcare, UK) as part of an AKTA 920 with 100 mM NaCl, 25 mM TrisCl, pH 8.0 running at 1 ml/min at room temperature. Peak fractions were

collected and concentrated first using a pre-rinsed Amicon Ultra-4 10 kDa column and then a Vivaspin 500 10 kDa column to roughly 100 μl, buffer-exchanged into 50 mM NaCl, 10 mM Tris, pH 8.0, concentrated to roughly 20 μl, and then transferred into a polypropylene microfuge tube. Acetic acid was added to 5% and the sample was concentrated to 2–4 μl using a SpeedVac at 40°C before analysis by MALDI-TOF (Protein and Nucleic Acid Chemistry services, Department of Biochemistry, University of Cambridge).

## Crystallography

Recombinant MHC class I complexes were crystallized using the sitting-drop method (see *Table 3* for conditions). Crystals were flash frozen in liquid nitrogen and native data sets for each crystal were collected at 100K [Diamond Light Source, Harwell (beamlines I02, I03, I04 or I04-1), or the ESRF, Grenoble (beamline ID29)]. Data were processed using either AUTOPROC (*Vonrhein et al., 2011*) or XIA2 (*Winter, 2010*) with XDS (*Kabsch, 2010*) for integration and AIMLESS (*Evans, 2006*) or SCALA (*Evans and McCoy, 2008*) for scaling. All structures were solved by molecular replacement, as implemented in Phaser (*McCoy et al., 2007*), part of the CCP4 software package (*Winn et al., 2011*). Starting molecular replacement models were generated using CHAINSAW (*Stein, 2008*) and the atomic co-ordinates of the chicken B21 MHC class I molecule (PDBID: 3BEV) with the peptide removed. Model building and refinement were carried out using COOT (*Emsley et al., 2010*) and AUTOBUSTER (*Bricogne et al., 2011*) or REFMAC (*Murshudov et al., 1997*), with the heavy and light chains of the MHC molecule rebuilt first before the peptide was modelled into residual electron density (see *Table 1* for refinement statistics, *Table 3* for data collection and refinement information, *Table 4* for Ramachandran statistics).

## Acknowledgements

Structure factors and atomic coordinates have been deposited with the Protein Data Bank under accession codes 2YEZ (BF2*2101 with TNPESKVFYL), 4D0B (BF2*2101 with TAGQEDYDRL), 4D0C (BF2*2101 with TAGQSNYDRL), 4CVZ (BF2*2101 with YELDEKFDRL), 4CVX (BF2*0201 with YPYLGPNTL), 4D0D (BF2*0201 with VIFPAKSL), and 4CW1 (BF2*1401 with SWFRKPMTR). We thank many colleagues at the IAH (now rebranded as the Pirbright Institute) for much historical help in processing materials for analysis of MHC-bound peptides, especially Larry Hunt from the Protein Chemistry facility for analysis of eluted peptides by Edman degradation and for provision of synthetic peptides.

We thank Diamond Light Source for access to beamlines I02, I03, I04, and I04-1 (mx9306) that contributed to the results presented here. The B14 structural experiments were performed on the ID29 beamline at the European Synchrotron Radiation Facility (ESRF), Grenoble, France. We are grateful to Local Contact at ESRF for providing assistance in using beamline ID29. We also thank Ed Lowe from the Biochemistry Department, Oxford for data collection and members of the Lea lab for crystallography discussions. We thank Steve Marsh from the Anthony Nolan, Tom Pizzari from Oxford Zoology, and Lars Råberg from Lund University for helpful discussions. We thank Gillian Griffiths, Alison Schuldt, Michaela Fakiola, and Mike Ratcliffe for critical reading of the manuscript. The authors declare no conflicts of interest.

## Additional information

### Funding

| Funder | Grant reference | Author |
| --- | --- | --- |
| Wellcome Trust | Programme grant 089305 | El Kahina Meziane, Michael Harrison, Łukasz Magiera, Clemens Hermann, Laura Mears, Antoni G Wrobel, Charlotte Durant, Jim Kaufman |
| Biotechnology and Biological Sciences Research Council (BBSRC) | Core Funding to the Pirbright Institute | William Mwangi, Colin Butter, Venugopal Nair |

| Funder | Grant reference | Author |
|---|---|---|
| Biotechnology and Biological Sciences Research Council (BBSRC) | PhD studentship | Paul E Chappell |
| Wellcome Trust | Senior Investigator Award | Susan M Lea |

The funders had no role in study design, data collection and interpretation, or the decision to submit the work for publication.

## Author contributions

PC, EKM, MH, NT, WM, Acquisition of data, Analysis and interpretation of data, Drafting or revising the article; ŁM, CH, LM, AGW, CD, PR, Acquisition of data, Analysis and interpretation of data; LLN, SB, Provided unique constructs and bacterial cells; CB, VN, Acquisition of data, Drafting or revising the article, Contributed unpublished essential data or reagents; TA, RD, AM, Acquisition of data, Drafting or revising the article; SML, Conception and design, Analysis and interpretation of data, Drafting or revising the article; JK, Conception and design, Acquisition of data, Analysis and interpretation of data, Drafting or revising the article, Contributed unpublished essential data or reagents

## Ethics

Human subjects: Anthony Nolan registrants signed written consent forms, with all procedures carried out under Human Tissue Act licensing number 22513 and with approval of the local Research Ethics committee (REC).

Animal experimentation: All procedures involving chickens were carried out at the University of Cambridge under Home Office project license PPL 80/2420 and with ethical approval of the Local Ethical Review Committee.

# Additional files

## Major datasets

The following datasets were generated:

| Author(s) | Year | Dataset title | Dataset ID and/or URL | Database, license, and accessibility information |
|---|---|---|---|---|
| Chappell PE, Roversi P, Harrison MC, Mears LE, Kaufman JF, Lea SM | 2011 | COMPLEX OF A B21 CHICKEN MHC CLASS I MOLECULE AND A 10MER CHICKEN PEPTIDE | http://www.rcsb.org/pdb/explore/explore.do?structureId=2YEZ | Publicly available at the RCSB Protein Data Bank 2YEZ. |
| Chappell PE, Roversi P, Harrison MC, Mears LE, Kaufman JF, Lea SM | 2014 | COMPLEX OF A B21 CHICKEN MHC CLASS I MOLECULE AND A 10MER CHICKEN PEPTIDE | http://www.rcsb.org/pdb/search/structidSearch.do?structureId=4D0B | Publicly available at the RCSB Protein Data Bank (4D0B). |
| Chappell PE, Roversi P, Harrison MC, Mears LE, Kaufman JF, Lea SM | 2014 | COMPLEX OF A B21 CHICKEN MHC CLASS I MOLECULE AND A 10MER CHICKEN PEPTIDE | http://www.rcsb.org/pdb/search/structidSearch.do?structureId=4D0C | Publicly available at the RCSB Protein Data Bank (4D0C). |
| Chappell PE, Roversi P, Harrison MC, Mears LE, Kaufman JF, Lea SM | 2014 | COMPLEX OF A B21 CHICKEN MHC CLASS I MOLECULE AND A 10MER CHICKEN PEPTIDE | http://www.rcsb.org/pdb/search/structidSearch.do?structureId=4CVZ | Publicly available at the RCSB Protein Data Bank (4CVZ). |
| Chappell PE, Roversi P, Harrison MC, Mears LE, Kaufman JF, Lea SM | 2014 | COMPLEX OF A B2 CHICKEN MHC CLASS I MOLECULE AND A 9MER CHICKEN PEPTIDE | http://www.rcsb.org/pdb/search/structidSearch.do?structureId=4CVX | Publicly available at the RCSB Protein Data Bank (4CVX). |
| Chappell PE, Roversi P, Harrison MC, Mears LE, Kaufman JF, Lea SM | 2014 | COMPLEX OF A B2 CHICKEN MHC CLASS I MOLECULE AND A 8MER CHICKEN PEPTIDE | http://www.rcsb.org/pdb/search/structidSearch.do?structureId=4D0D | Publicly available at the RCSB Protein Data Bank (4D0D). |

| Author(s) | Year | Dataset title | Dataset ID and/or URL | Database, license, and accessibility information |
|---|---|---|---|---|
| Chappell PE, Roversi P, Harrison MC, Mears LE, Kaufman JF, Lea SM | 2014 | COMPLEX OF A B14 CHICKEN MHC CLASS I MOLECULE AND A 9MER CHICKEN PEPTIDE | http://www.rcsb.org/pdb/search/structidSearch.do?structureId=4CW1 | Publicly available at the RCSB Protein Data Bank (4CW1). |

The following previously published datasets were used:

| Author(s) | Year | Dataset title | Dataset ID and/or URL | Database, license, and accessibility information |
|---|---|---|---|---|
| Koch M, Kaufman J, Jones Y | 2007 | 11mer Structure of an MHC class I molecule from B21 chickens illustrate promiscuous peptide binding | http://www.rcsb.org/pdb/explore/explore.do?structureId=3BEV | Publicly available at the RCSB Protein Data Bank (3BEV). |
| Koch M, Kaufman J, Jones Y | 2007 | 10mer Crystal Structure of chicken MHC class I haplotype B21 | http://www.rcsb.org/pdb/explore/explore.do?structureId=3BEW | Publicly available at the RCSB Protein Data Bank (3BEW). |

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
