## [Decision Letter]

Thank you for sending your work entitled “Expression Levels of MHC Class I molecules are Inversely Correlated with Promiscuity of Peptide Binding” for consideration at *eLife*. Your article has been favorably evaluated by Tadatsugu Taniguchi (Senior editor), a Reviewing editor, and three reviewers, one of whom, Andrej Kosmrlj, has agreed to reveal his identity.

The Reviewing editor and the reviewers discussed their comments before we reached this decision, and the Reviewing editor has assembled the following comments to help you prepare a revised submission.

All reviewers agree that this paper may represent an important study. You aim to demonstrate both in chickens and humans that the expression levels of MHC molecules on the surface of cells are inversely correlated with promiscuity of peptide binding. Interestingly MHC types, which are promiscuous for peptide binding and are thus expressed at lower levels, are associated with resistance to Marek's disease in chickens, while in humans they are associated with faster progression to AIDS. The opposite is true for MHC types, which are more restrictive for peptide binding. However, all reviewers also agree that several important points need to be addressed, and some speculations refined, before this paper is acceptable for publication. The points that need to be addressed are summarized below:

1) The authors state in the Results section, that both antibodies used to measure HLA*B expression levels gave similar but non-identical results, which suggested to them that they recognize different epitopes on the HLA-B molecules. This is very likely to be an incorrect conclusion, as [4] present data strongly indicating that these antibodies actually do recognize the same epitope. The possibility that expression levels of HLA-B correlate with peptide binding promiscuity in humans is very intriguing, but the authors are encouraged to expand their monoclonals to a set that will more definitively measure HLA-B allotype expression. There are antibodies that recognize certain HLA-B allotypes specifically and with equal affinity, including pairs of the allotypes included in Figure 7. The bridging of this work from chickens (where the correlation data between peptide binding promiscuity and expression levels are clear cut) to humans is a major point of the paper, so the expression data need to be definitive. Finally, if possible, inclusion of data on HLA-A would strengthen conclusions drawn from Figure 7?

2) Repertoire size in the study has been defined on the basis of the breadth of the primary anchor motif. But, it is not clear that this is the same as the number of peptides actually bound and presented. An allele with a diverse motif, defined here as having a large repertoire, might in the end present fewer peptides because nothing really binds well. Conversely, an allele with a narrow motif might bind everything that has the motif, and in the end have a large repertoire. In this respect, defining repertoire size on the basis of motif diversity, and not on the basis of actually bound peptides, is a limitation. Some way to quantify repertoire size beyond just the number of amino acids preferred at the various anchor residues needs to be established.

3) The demarcation of alleles into generalist and specialists is interesting. However, this is likely to be a continuum, and not a strict dichotomy. At the same time, surface expression is also likely to be a continuum. The present study suggests that these two continuums run in parallel (but opposite directions). There is too little data to convincingly establish this. An example of a manifestation of the looseness of this argument is the following. According to the authors, the HLA-B57 allele would be classified as a specialist for a particular pathogen. However, it is known that they are associated with protection against HIV and hepatitis C virus (HCV). On the other hand the more promiscuous allele HLA-B8 would be classified as a generalist, but this allele is known to be associated with faster disease progression in HIV and HCV. In this regard, it may be good to also list T cell cross-reactivity as a potentially significant variable for highly mutable pathogens (see Kosmrlj et al), and the importance of presenting peptides from mutationally vulnerable regions of HIV (see Dahirel et al, 2011), Ferguson et al, 2013, and other references). What can be said about other infections where there is significant HLA association?

4) The authors suggest that the difference in cell surface expression level in chickens might have to do with peptide loading and editing? Is there any data that suggest peptide loading affects surface expression levels? This would be important to cite.

5) Is there any data to suggest that lower expression levels of class I leads to escape from negative selection (which is implied in the Discussion section, paragraph eight)?

6) The Discussion is thought-provoking, but a bit long and could do with some tightening and perhaps less speculation in cases where background data for the models do not yet exist.

---

## [Author Response]

*All reviewers agree that this paper may represent an important study. You aim to demonstrate both in chickens and humans that the expression levels of MHC molecules on the surface of cells are inversely correlated with promiscuity of peptide binding. Interestingly MHC types, which are promiscuous for peptide binding and are thus expressed at lower levels, are associated with resistance to Marek's disease in chickens, while in humans they are associated with faster progression to AIDS. The opposite is true for MHC types, which are more restrictive for peptide binding. However, all reviewers also agree that several important points need to be addressed, and some speculations refined, before this paper is acceptable for publication. The points that need to be addressed are summarized below*:

*1) The authors state in the Results section that both antibodies used to measure HLA*B expression levels gave similar but non-identical results, which suggested to them that they recognize different epitopes on the HLA-B molecules. This is very likely to be an incorrect conclusion, as*
[4]
*present data strongly indicating that these antibodies actually do recognize the same epitope. The possibility that expression levels of HLA-B correlate with peptide binding promiscuity in humans is very intriguing, but the authors are encouraged to expand their monoclonals to a set that will more definitively measure HLA-B allotype expression. There are antibodies that recognize certain HLA-B allotypes specifically and with equal affinity, including pairs of the allotypes included in*
Figure 7*. The bridging of this work from chickens (where the correlation data between peptide binding promiscuity and expression levels are clear cut) to humans is a major point of the paper, so the expression data need to be definitive. Finally, if possible, inclusion of data on HLA-A would strengthen conclusions drawn from*
Figure 7*?*

We are glad that, in this part of the review, the reviewers consider the work with chicken class I molecules to be definitive, and we share their desire to see even more data for human class I molecules. To put this request into perspective, the two papers on the expression level of human class I molecules (Nature Genetics and Science, for which the level of proof should have been high) describe data with one single monoclonal antibody directed to HLA-C and large panels of humans bearing all kinds of HLA types, mostly heterozygote for HLA-C, giving a vast range of expression levels from which the authors derive a statistical result. Inspection of the data shows that the concordance of alleles with expression level is not perfect, even for identified homozygotes.

We chose a different approach, which was to pick individuals examined in great detail for their MHC alleles that had the right array of class I alleles to allow us to examine the four HLA-B alleles without confounding effects of heterozygosity or HLA-A cross-reaction. It took us over two years to get all these data, in large part because these carefully chosen individuals are only willing to donate at times of their own convenience. As appealing as it is (for us as well as for the reviewers) to gather substantially more data, recruitment of new donors and subsequent experiments are likely to take us at least another two years, and therefore are outside the scope of this current paper.

We also chose to use two different monoclonal antibodies, twice as many as were used in the two previous publications. On the basis of the similar but non-identical results of the two antibodies with the four kinds of donors, we suggested that the epitopes for these antibodies are likely not to be identical. The reviewer countered that data in the [5] paper suggests that the epitopes are identical, presumably (since they did not state why) because of concordance of reaction with a range of class I alleles, with the implication that the two antibodies were not independent and thus were no better than previous data with HLA-C. We have responded in two ways. First, we test whether either antibody will inhibit the other, finding that the two antibodies do inhibit each other and therefore have at the least overlapping epitopes, but that one inhibited the other much better than the reverse suggesting that they have very different affinities. Second, we used a third monoclonal antibody that recognizes only HLA-B57 and HLA-B27 (and thus is extremely likely to recognize a separate epitope) and found the same pattern with two donors as with the other two antibodies.

Finally, the reviewers ask us to examine many other HLA allele pairs with monoclonal antibodies. We chose the four HLA-B alleles because they have been examined in the [29], Nature article (for which, again, the level of proof should have been high) that had gone some way towards understanding the story we have been working on for 20 years. It took us two years to recruit the donors for the experiments involving two monoclonal antibodies with these four alleles, and a further two months to recruit two further donors to examine with an additional antibody in response to the review. A study with the many antibodies envisaged by the reviewers would be extremely interesting, but it likely to take years to complete in a worthwhile fashion, and therefore is outside the scope of the current paper.

*2) Repertoire size in the study has been defined on the basis of the breadth of the primary anchor motif. But, it is not clear that this is the same as the number of peptides actually bound and presented. An allele with a diverse motif, defined here as having a large repertoire, might in the end present fewer peptides because nothing really binds well. Conversely, an allele with a narrow motif might bind everything that has the motif, and in the end have a large repertoire. In this respect, defining repertoire size on the basis of motif diversity, and not on the basis of actually bound peptides, is a limitation. Some way to quantify repertoire size beyond just the number of amino acids preferred at the various anchor residues needs to be established*.

This is an important point that we should have considered. As a result, we now include an experiment comparing B19 and B21 cell lines. The B19 cells have twice as many class I molecules on the cell surface as assessed by flow cytometry, but have only one third as many sequences of peptides as assessed by immuno-isolation followed by mass spectrometry analysis. These results show that promiscuous class I alleles can and do bind a greater variety of peptides that are presented on the cell surface.

*3) The demarcation of alleles into generalist and specialists is interesting. However, this is likely to be a continuum, and not a strict dichotomy. At the same time, surface expression is also likely to be a continuum. The present study suggests that these two continuums run in parallel (but opposite directions). There is too little data to convincingly establish this*.

This portion of the comment appears to repeat our arguments back to us. In the Discussion we say “the work presented in this paper has begun to define two groups (or a range between two extremes) of class I molecules that are strategically evolved for different modes of resistance”, although due to space limitations we only mention the two groups in the Abstract. Of course more data is always welcome, but we judge that we have shown enough data to support our argument.

*An example of a manifestation of the looseness of this argument is the following. According to the authors, the HLA-B57 allele would be classified as a specialist for a particular pathogen. However, it is known that they are associated with protection against HIV and hepatitis C virus (HCV). On the other hand the more promiscuous allele HLA-B8 would be classified as a generalist, but this allele is known to be associated with faster disease progression in HIV and HCV. In this regard, it may be good to also list T cell cross-reactivity as a potentially significant variable for highly mutable pathogens (see Kosmrlj et al), and the importance of presenting peptides from mutationally vulnerable regions of HIV (see Dahirel et al, 2011*, *Ferguson et al, 2013, and other references). What can be said about other infections where there is significant HLA association?*

We are glad that reviewers find the generalist and specialist idea interesting, but it is only one of several ways we propose as ways to look at this phenomenon, and we are careful to say that the different proposals are not necessarily mutually exclusive. In the Discussion we state: “One possible strategy is that fastidious class I molecules with pauci-clonal T cell responses might be better suited for rapidly evolving viruses with a limited scope for immune evasion, for which a particular peptide might be the most efficient way to achieve resistance. In contrast, promiscuous class I molecules might be better suited for pathogens with coding potential for many immune evasion genes and more stability over evolutionary time, such as large DNA viruses, bacteria and even parasites.” The fact that B57 protects from both HIV and HCV, while B8 does not, may lend support to the proposal that fastidious class I molecules are better for viruses with mutable regions while promiscuous molecules are better for more stable pathogens with larger genomes. However, such an analysis as suggested is well beyond the scope of the present paper.

*4) The authors suggest that the difference in cell surface expression level in chickens might have to do with peptide loading and editing? Is there any data that suggest peptide loading affects surface expression levels? This would be important to cite*.

The first sentence is phrased as a question, and there is no doubt that we have made this point as a proposal in the Discussion. To our knowledge, there is little pertinent data published even for mammals; we already have cited [44], J Immunol.

5) Is there any data to suggest that lower expression levels of class I leads to escape from negative selection (which is implied in the Discussion section, paragraph eight)?

We are not aware of any published data on this point, but it is stated explicitly (not implied as asserted by the reviewer) as part of our proposal in the Discussion that flows directly from our interpretation of the data presented in this manuscript and is an ongoing area of research in our laboratory. Extension of data into novel hypotheses is standard scientific practice, as a way of stimulating interest in further experimentation. For instance, in their 2010 Nature article Kosmrlj et al proposed that promiscuous class I molecules led to pauperized T cell repertoires, a proposal for which they presented no experimental evidence but which has stimulated further thought (not the least for us).

*6) The Discussion is thought-provoking, but a bit long and could do with some tightening and perhaps less speculation in cases where background data for the models do not yet exist*.

We have made changes to the Discussion to clarify and add support to the proposals made. In any case, our main text is within 150 words of the 5000 words recommended by the current Author Guides and Policies published on-line by *eLife*. Therefore, we see no reason to shorten or limit the Discussion, which we hope is full of interesting new ideas.